# Mapping the Distribution and Dispersal Risks of the Alien Invasive Plant *Ageratina adenophora* in China

**Xiaojuan Zhang** [1,2], **Yanru Wang** [1,2], **Peihao Peng** [1,2,3,*], **Guoyan Wang** [2,3], **Guanyue Zhao** [1,2], **Yongxiu Zhou** [4] **and Zihao Tang** [5]

1. College of Earth Sciences, Chengdu University of Technology, Chengdu 610059, China
2. Institute of Ecological Resources and Landscape Architecture, Chengdu University of Technology, Chengdu 610059, China
3. College of Tourism and Urban-Rural Planning, Chengdu University of Technology, Chengdu 610059, China
4. College of Geophysics, Chengdu University of Technology, Chengdu 610059, China
5. Chengdu Academy of Agriculture and Forestry Science, Chengdu 611130, China
* Correspondence: pengpeihao@cdut.edu.cn

**Abstract:** Identifying the distribution dynamics of invasive alien species can help in the early detection of and rapid response to these invasive species in newly invaded sites. *Ageratina adenophora*, a worldwide invasive plant, has spread rapidly since its invasion in China in the 1940s, causing serious damage to the local socioeconomic and ecological environment. To better control the spread of this invasive plant, we used the MaxEnt model and ArcGIS based on field survey data and online databases to simulate and predict the spatial and temporal distribution patterns and risk areas for the spread of this species in China, and thus examined the key factors responsible for this weed's spread. The results showed that the risk areas for the invasion of *A. adenophora* in the current period were 18.394° N–33.653° N and 91.099° E–121.756° E, mainly in the tropical and subtropical regions of China, and densely distributed along rivers and well-developed roads. The high-risk areas are mainly located in the basins of the Lancang, Jinsha, Yalong, and Anning Rivers. With global climate change, the trend of continued invasion of *A. adenophora* is more evident, with further expansion of the dispersal zone towards the northeast and coastal areas in all climatic scenarios, and a slight contraction in the Yunnan–Guizhou plateau. Temperature, precipitation, altitude, and human activity are key factors in shaping the distribution pattern of *A. adenophora*. This weed prefers to grow in warm and precipitation-rich environments such as plains, hills, and mountains; in addition, increasing human activities provide more opportunities for its invasion, and well-developed water systems and roads can facilitate its spread. Measures should be taken to prevent its spread into these risk areas.

**Keywords:** alien invasive plants; *Ageratina adenophora*; species distribution models; driving factors; dispersal risk; climate change

## 1. Introduction

Since Elton published *The ecology of invasions by animals and plants*, invasive alien species have been the focus of attention in the field of ecology [1]. China is one of the countries most seriously affected by invasive alien species in the world. The economic loss caused by alien species to the Chinese economy and environment has exceeded CNY 200 billion [2]. According to the 2020 China Ecological Environment Status Bulletin, more than 660 invasive alien species have been discovered across the country, and 71 of them have posed or are potential threats to natural ecosystems and have been added to China's list of invasive alien species. Among the 660 invasive alien species, there are 370 invasive plants, 220 invasive animals, and 60 invasive microorganisms. Therefore, it is urgent and necessary to carry out research on preventing biological invasion [3].

Invasive alien species are defined as the spread of nonindigenous species from their original place to a new area, where it colonizes and spreads further and poses a threat

to the local economy, environment, and ecosystem health [4–7]. Invasive alien species threaten the global sustainability of the biodiversity and social economy by altering the ecosystem structure and functioning and disrupting key biological interactions [8–12]. These invasions have also been considered a major cause of recent extinctions and an important component of global environmental change [13]. Plant invasions, as an important component of invasive alien species, have been a major research topic for ecologists in recent years [14–17]. *Ageratina adenophora* (Spreng.) King and H. Rob is regarded as one of the most serious invasive species in most countries in the world [18]. This perennial semi-shrubby herb of the genus *Ageratina* (Asteraceae) is native to Mexico and Costa Rica. It was introduced to Yunnan Province of China from the Myanmar border in the 1940s and quickly spread throughout southwestern China [19]. As one of the earliest invasive alien species in China, the *A. adenophora* has crowded out native species in its invasive range due to the strong reproductive ability of the plant and the allelopathic effects of the root system in the soil, mostly in single dominant species distributed in dense patches [20]; therefore, it is difficult to eradicate completely. To make matters worse, the weed has shown great potential to spread outwards and has become widespread in southwestern China, causing huge losses to local agriculture, forestry, and livestock [21,22]. Therefore, it is urgent and necessary to carry out research on the invasion of *A. adenophora* in China. The control of biological invasions is the ultimate goal of biological invasion research [23–25]; it is generally accepted that the prevention of invasive plant outbreaks before they occur is a better economic strategy than control or eradication afterward [26,27]. Therefore, predicting the risk areas for the spread of *A. adenophora* before its colonization can effectively control the spread of this invasive plant.

In early detection, the use of species distribution models (SDMs) is effective in identifying ecologically sensitive areas and in monitoring and rapidly responding to invasive species [28,29]. A key assumption for using the species distribution model is the niche conservatism hypothesis, which assumes that species largely maintain their niches in space and time: niches will change slowly, so species can only sustain their populations under similar conditions [30,31]. However, some studies have found evidence of rapid niche changes across space and time, sparking widespread debate about whether SDMs can be transferred to new areas or time periods. In order to examine how niche change affects the transferability of species distribution models, a recent study by Liu et al. (2022) has proved that species occupying similar niches between the native and introduced areas will have a high SDM transferability [32]. SDMs correlate information on the spatial distribution of species with environmental variables to produce a model that expresses the realized niche of a species, and the model can be projected to different geographic areas or times to predict the geographical distribution patterns of species [33,34]. The MaxEnt model is a machine learning model based on the MaxEnt algorithm developed by S.J. Phillips et al., in 2006; it only uses species "presence" data to build models [35]. In many datasets that lack information about species absence and small sample sizes, the MaxEnt model performs better than other models [36,37]. Because of its realistic species habitat simulations, screening of major ecological environmental factors, and quantitative descriptions of environmental factors in species habitats, the MaxEnt model has been widely used in a variety of disciplines, including geography and ecology, for species distribution prediction analysis since 2004 [38–40].

Climatic factors are considered to be the main environmental drivers of species distribution at the macro-scale, and this has been established in extensive studies of the dispersal distribution of invasive species [41–43]. However, there are still many important factors that need to be considered when predicting the potential range of invasive alien species, such as topography, soil type, land use/cover change, and human activity. Topographical factors can play a role in redistributing solar radiation and precipitation, and soil factors can affect the growth of invasive plant roots and nutrient uptake [44,45]. Land use/cover change and anthropogenic factors affect the colonization and spread of invasive plants, mainly by disrupting their habitat [46–49]. For future climate scenarios, most previous studies were based on the coupled model intercomparison project phase 5 (CMIP5), while

the new version CMIP6 has shown significantly higher accuracy and resolution [50]. In a comparison of the performance of the CMIP6 and CMIP5 models in simulating historical precipitation and temperature in Bangladesh, Kamruzzaman et al., found that the CMIP6 MME showed a significant improvement in simulating rainfall and temperature over Bangladesh compared to CMIP5 MME [51]. Chen et al., showed that the new version of the CMIP6 model exhibits a general improvement in the simulation of climate extremes and their trend patterns compared to the old version of the model in CMIP5, and the simulation results are closer to the observations [52]. In addition, an increasing number of researchers have recently used CMIP6 models for species distribution prediction [53,54]. As a result, additional consideration of topography, soils, land use, and anthropogenic factors under CMIP6 is expected to result in a more reasonable possibility of predicting the distribution of invasive alien plants.

We used the MaxEnt model and ArcGIS software to simulate and predict the current spatial and temporal distribution and dispersal risk areas of *A. adenophora* based on occurrence records and environmental data. The goals of this study were to (1) map the current geographical distribution of *A. adenophora*, (2) reveal the key drivers affecting the weed distribution, and (3) forecast the spreading trend of weeds in China. These findings will provide predictive, early warning information for invasive plant prevention and control.

## 2. Materials and Methods

### 2.1. Species Occurrence Records

We obtained the species distribution information of *A. adenophora* from the Global Biodiversity Information Facility (GBIF, https://www.gbif.org) (accessed on 28 April 2022), the Chinese Virtual Herbarium (CVH, https://www.cvh.ac.cn/) (accessed on 28 April 2022), the China National Specimen Information Infrastructure (NSII, http://www.nsii.org.cn/2017/home.php) (accessed on 28 April 2022), the related literature, and our field survey data. To prevent spatial autocorrelation of the raster data, we used ArcGIS10.6 (ESRI) to ensure that only one occurrence of this species was recorded in each 1 km × 1 km resolution grid cell [55]. *A. adenophora* occurs between 92.273° E–120.798° E and 21.496° N–29.824° N and is currently distributed mainly in southwestern China, including Yunnan, Sichuan, Chongqing, Tibet, Guizhou, Guangxi, and Taiwan (Figure 1).

### 2.2. Environmental Data and Pre-Processing

We downloaded 19 grid-based bioclimatic variables from the WorldClim (www.worldclim.org) (accessed on 19 May 2022) dataset [56]; the spatial resolution was 30″ (approximately 1 km). Elevation, land use type, soil type, and soil texture were downloaded from the Resource and Environmental Science and Data Centre (http://www.resdc.cn/) (accessed on 25 May 2022) at a resolution of approximately 1 km [57]; we extracted the slope and aspect from elevation based on the ArcGIS10.6 platform. The seven variables related to soil quality were taken from the Harmonized World Soil Database v1.2 dataset of the World Food and Agriculture Organization (http://www.fao.org/soils-portal/soil-survey/soil-maps-and-databases/harmonized-world-soil-database-v12/en/) (accessed on 21 May 2022) [58,59], with a resolution of approximately 30″. The anthropogenic intensity data were obtained from the Global Human Influence Index (Geographic), v2 (1995–2004) dataset of the Socioeconomic Data and Applications Center (SEDAC, https://sedac.ciesin.columbia.edu/data/set/wildareas-v2-human-influence-index-geographic) (accessed on 21 May 2020) at a resolution of approximately 1 km. The spatial resolution of all environmental data was set to 30″. A list of environmental data is available in the Supplementary Material (Table S1). Based on the condition that soil and topographic factors remain constant over the next few decades, only climate variables from future climate scenarios were used in this study to predict the dispersal trends of *A. adenophora* over future periods.

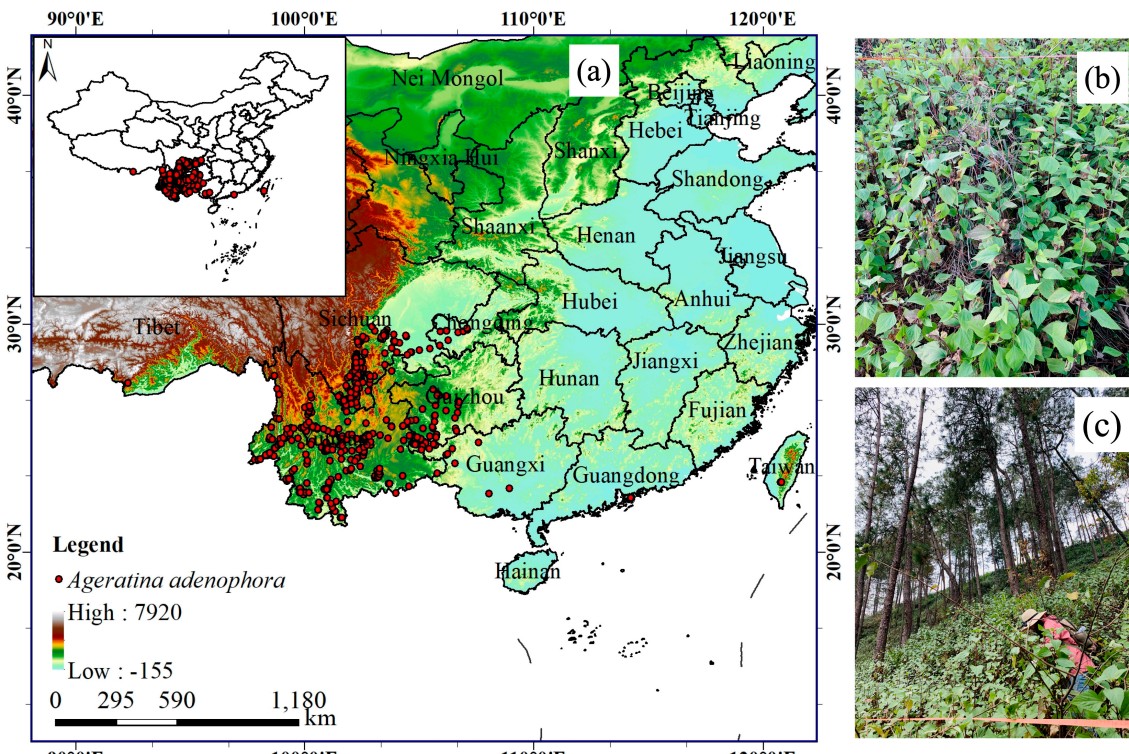

**Figure 1.** The map (**a**) shows the occurrence records of *Ageratina adenophora* in China; (**b**,**c**) show *Ageratina adenophora*.

Environmental variables are important parameters in the construction of species distribution models, and too much co-linearity between environmental variables can cause over-fitting of the model and lead to reduced model transfer ability [60,61]. To ensure that all environmental predictor variables were independent of each other, we tested the correlation of 35 variables using the PerformanceAnalytics package [62] in R and removed those with high spatial correlation values (Spearman correlation > 0.75) [63]. Finally, we selected eighteen variables as predictor variables, including mean diurnal range, temperature annual range, mean temperature of driest quarter, annual precipitation, precipitation of driest quarter, altitude, slope, aspect, nutrient availability, nutrient retention capacity, rooting conditions, excess salts, soil type, soil texture (clay, sand, silt), land use type (agricultural land, forest land, grassland, water, building land, unused land), and intensity of human activity (Table 1).

We selected the BCC-CSM2-MR global climate model from the coupled phase 6 (CMIP6) model for our projections because of its higher resolution and climate sensitivity compared to CMIP5 [64,65], and the better performance of the BCC-CSM2-MR model compared to other global climate models [66]. For the four shared socioeconomic pathways (SSP126, SSP245, SSP370, and SSP585), we chose SSP126 for the optimistic scenario and SSP 585 for the pessimistic scenario for model simulation under the future climate scenario, as they combine factors affecting local development and a higher climate sensitivity [67]. In addition, two periods were chosen to predict the spread of the invasive alien plant *A. adenophora*: the 2050s (predicted mean for 2041–2060) and the 2090s (predicted mean for 2081–2100).

**Table 1.** The environmental variables for potential risk modeling of *Ageratina adenophora* in China.

| Environment Variables | Description | Abbreviation |
|---|---|---|
| Climate | Mean diurnal range (mean of monthly (max temp–min temp)) (0.1 °C) | bio_2 |
| | Temperature annual range (0.1 °C) | bio_7 |
| | Mean temperature of driest quarter (0.1 °C) | bio_9 |
| | Annual precipitation (mm) | bio_12 |
| | Precipitation of driest quarter (mm) | bio_17 |
| Soil | Soil type | soil |
| Soil texture | Sand | sand |
| | Silt | silt |
| | Soil | clay |
| Soil quality | Nutrient availability | sq1 |
| | Nutrient retention capacity | sq2 |
| | Rooting conditions | sq3 |
| | Excess salts | sq5 |
| | Workability (constraining field management) | sq7 |
| Land use type | Land use/cover change | lucc |
| Human activity | Human footprint dataset | hii |
| Terrain | DEM | dem |
| | Slope | slope |
| | Aspect | aspect |

### 2.3. Model Construction and Data Processing

Our experiments were conducted using MaxEnt 3.4.4 software for model construction (http://biodiversityinformatics.amnh.org) (accessed on 1 March 2022). The *A. adenophora* distribution points and current environmental variables were imported into the software, the output type was set to Logistic, 25% of the species distribution data was selected as the test set and the remaining 75% was used as the training set, and the Jackknife measure was enabled to analyze the model factor contribution. Due to the uncertainty in the species distribution model, the model was run for 10 iterations with the type of operation set to cross-validation. All other parameters were left at their default values and the final output was averaged over 10 iterations.

### 2.4. Niche Similarity Examination

*A. adenophora* was introduced to China from Myanmar around 80 years ago (1940s), and it quickly colonized and spread in southwest China. According to previous research, *A. adenophora*'s climate niche has slightly expanded in time and space since its invasion into China, but it still maintains high stability [68]. Furthermore, we compared the climate niches of *A. adenophora* between its native (Mexico) and introduced area (China) (Tables 2 and 3) and found that the first five environmental factors (dem, bio_7, bio_12, hii, and bio_9) in these two places are the same, indicating temperature, precipitation, altitude, and human activities in these two places have important contributions to the distribution of *A. adenophora*. These results demonstrated that our model is general for this species.

**Table 2.** The percent contribution and permutation importance of the environmental variables in Mexico.

| Variable | Percent Contribution (%) | Permutation Importance (%) |
|---|---|---|
| dem | 32.8 | 15.4 |
| bio_7 | 27.5 | 28 |
| bio_12 | 12.6 | 26.3 |
| hii | 9.1 | 5.2 |
| bio_9 | 4.2 | 5.1 |
| bio_3 | 4.2 | 2.3 |
| bio_19 | 2.7 | 7.3 |
| slope | 2.1 | 1.3 |
| soil | 1.4 | 0.4 |
| aspect | 1.3 | 1.1 |
| sq3 | 0.5 | 1.6 |
| sq4 | 0.4 | 1.4 |
| sq1 | 0.4 | 0.3 |
| bio_17 | 0.4 | 2.5 |
| sq5 | 0.3 | 1.7 |

**Table 3.** The percent contribution and permutation importance of the environmental variables in China.

| Variable | Percent Contribution (%) | Permutation Importance (%) |
|---|---|---|
| bio_7 | 27.3 | 33.3 |
| dem | 22.5 | 14.8 |
| bio_12 | 20.4 | 0.2 |
| bio_9 | 16.4 | 37.9 |
| hii | 7.8 | 5.7 |
| bio_2 | 1.7 | 0.1 |
| sand | 0.7 | 0.4 |
| clay | 0.6 | 0.6 |
| bio_17 | 0.5 | 3 |
| aspect | 0.4 | 0.5 |
| sq2 | 0.4 | 0.6 |
| soil | 0.3 | 0 |
| silt | 0.3 | 1.4 |
| sq1 | 0.2 | 0.3 |
| sq3 | 0.1 | 0.6 |
| lucc | 0.1 | 0.2 |
| slope | 0.1 | 0.2 |
| sq5 | 0.1 | 0.3 |

## 3. Results

### 3.1. Model Accuracy Assessment

The MaxEnt species distribution model was tested for accuracy by the area under the curve (AUC) of the receiver operating characteristic (ROC), with the AUC ranging between 0.5 and 1.0. The closer the AUC is to 1.0, the higher the accuracy of the model; if the AUC is equal to 0.5, the model has no application value [39]. The mean AUC value of the MaxEnt model for 10 operations in the study was 0.980 (Figure 2), with a standard deviation of 0.007, and the AUC values for a single operation were all larger than 0.9, indicating that the model performed well in terms of accuracy and could reflect the distribution characteristics of *A. adenophora*.

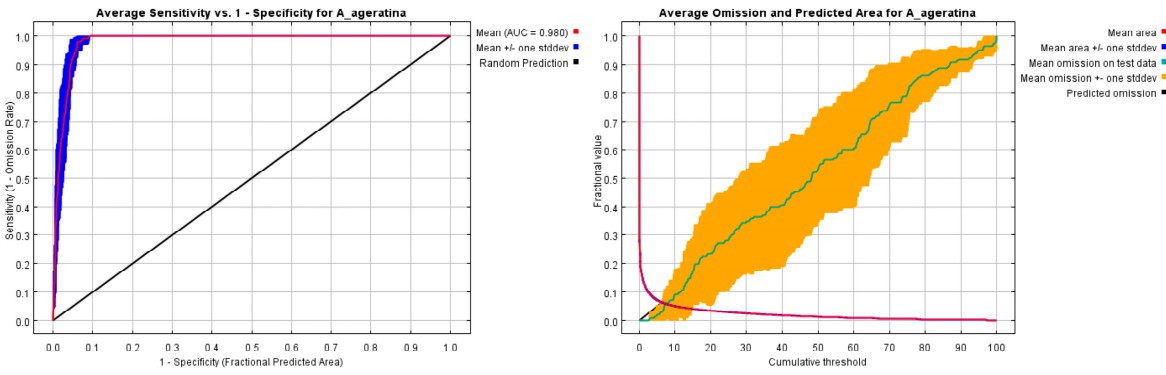

**Figure 2.** ROC curve and AUC value under current climate conditions.

### 3.2. Key Drivers Affecting the Distribution of A. adenophora

The factor contribution is an important indicator to assess the degree of influence of environmental variables on the model (Figure 3; Table 3). Of the 18 environmental variables, temperature annual range (bio_7), elevation (dem), mean annual precipitation (bio_12), mean temperature of driest quarter (bio_9), and human activity (hii) are the key drivers affecting the distribution of *A. adenophora*. The cumulative values for percentage contribution and permutation importance were as high as 94.4% and 91.9%, respectively. Of these, temperature annual range (bio_7) was the largest influencing factor, while mean temperature of driest quarter (bio_9) had the highest permutation importance (Table 3). Factors related to temperature variables (e.g., bio_7, bio_9) and dem variables had higher weights when only a single variable was used. However, soil variables (e.g., soil quality, soil type, soil texture) and land use/cover changes had negligible effects on the distribution of this weed. Their cumulative contribution and cumulative permutation importance are both less than 10%.

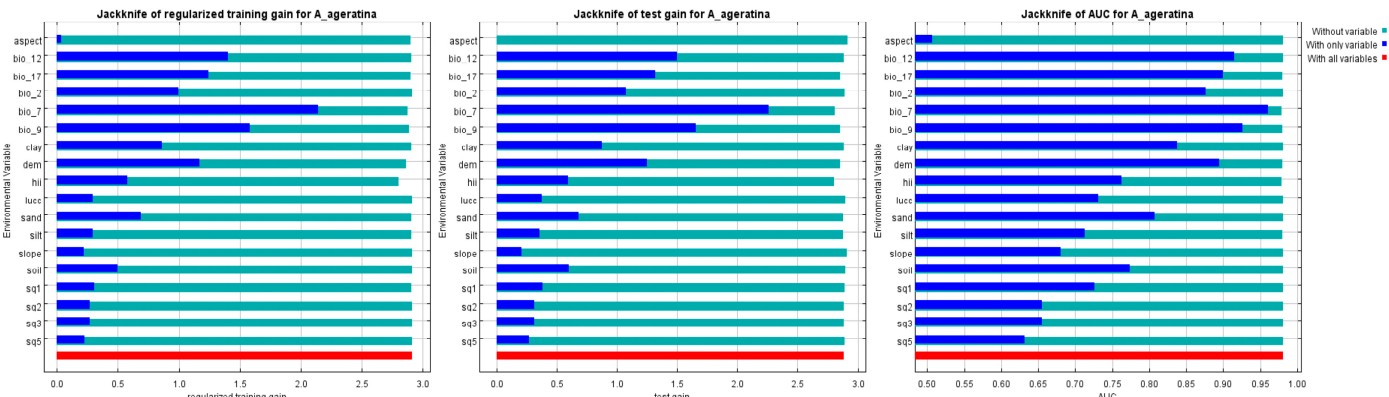

**Figure 3.** Jackknife test results of the MaxEnt model.

### 3.3. Characteristics of the Potential Risk Area of the A. Adenophora under Current Climate Conditions

Based on the results of the MaxEnt model simulations, we used the maximum training sensitivity plus specificity threshold (MTSS) to map the current and future invasion risk of the exotic plants. We classified the dispersal areas of *A. adenophora* into four categories based on the MTSS values corresponding to: high-risk areas (0.5 < MTSS), moderate-risk areas (0.25 < MTSS ≤ 0.5), low-risk areas (MTSS ≤ 0.25), and no-risk areas (0 < MTSS).

Our results show that the current invasion-risk area of *A. adenophora* reaches about $56.11 \times 10^4$ km², which is distributed in the tropical and subtropical regions of China, with a more concentrated distribution in the Yunnan, Sichuan, and Guizhou Provinces (Table 4; Figure 4). The simulating results are consistent with the actual distribution range of *A. adenophora*. The high-invasion-risk area is about $9.30 \times 10^4$ km², accounting for 16.58% of the total invasion area, and it is mainly distributed in areas with sufficient water sources and well-developed road networks (Table 4; Figure 4). The western part is invaded along the Lancang, Nujiang, Jinsha, and Yalong Rivers in many places, while the eastern part is concentrated in the Yunnan–Guizhou plateau. The moderate-invasion-risk area covers about $21.40 \times 10^4$ km², accounting for 38.13% of the total invasion area, and it is mainly distributed around the high-invasion-risk area and also in the Yunnan–Guizhou Plateau and Taiwan. The low-invasion-risk zone covers about $25.41 \times 10^4$ km², accounting for 45.29% of the total invasion zone; the *A. adenophora* mainly grows at the edge of the high- and moderate-invasion-risk zones, and is also partially distributed in the Yangtze River basin, the lower reaches of the Yarlung Tsangpo River, and in Taiwan, Guangdong, Guangxi, and Fujian Province.

**Table 4.** Dynamics of *Ageratina adenophora* invasion-risk areas at different times.

| | Climate Scenarios | Area (×10⁴ km²) | | | | Area Ratio (%) | | |
|---|---|---|---|---|---|---|---|---|
| | | Low Risk | Moderate Risk | High Risk | Total Area | Low Risk | Moderate Risk | High Risk |
| Current | - | 25.41 | 21.40 | 9.30 | 56.11 | 45.29 | 38.13 | 16.58 |
| 2050s | ssp126 | 24.66 | 22.05 | 8.83 | 55.53 | 44.40 | 39.70 | 15.90 |
| | ssp585 | 27.36 | 24.38 | 8.76 | 60.50 | 45.22 | 40.31 | 14.48 |
| 2090s | ssp126 | 31.33 | 21.09 | 9.27 | 61.69 | 50.79 | 34.19 | 15.02 |
| | ssp585 | 28.77 | 21.90 | 7.43 | 58.10 | 49.52 | 37.70 | 12.78 |

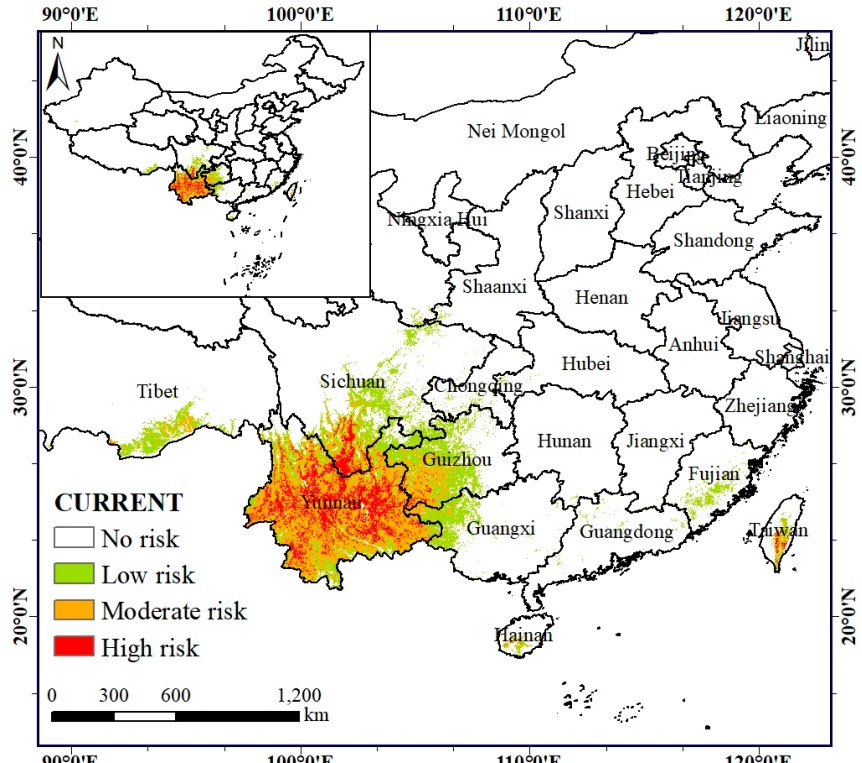

**Figure 4.** Potential risk area of the *Ageratina adenophora* under current climate conditions.

### 3.4. Evaluation of the Dispersal Dynamics of A. adenophora

The predicted results show a general trend of an increasing area of the *A. adenophora* invasion-risk zone between the current time and the 2090s, with a spatial expansion mainly towards the northern and coastal regions. The current and future climatic scenarios show that the area of low-risk invasion zones for this weed is the largest, followed by the area of moderate-risk zones, and the area of high-risk zones is the smallest (Table 4; Figure 5).

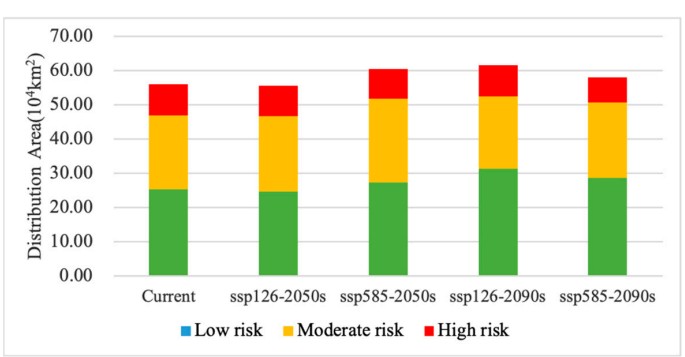

**Figure 5.** Dynamics of *Ageratina adenophora* invasion-risk areas.

The results show an overall increasing trend in the area of *A. adenophora* under the future SSP126 climatic scenario, with a slight decrease of 1.02% by the 2050s. The weed will invade in the northern, northeastern, and coastal areas, especially along the Jialing River, the Yangtze River, and coastal areas such as Guangzhou, Fujian, and Taiwan, where the area of spread is increasing more (Figures 6 and 7). The invasion-risk area of *A. adenophora* increased significantly in the 2090s compared to the 2050s, with the largest increase of 6.39% in the low-risk area, which was primarily distributed around the Yunnan–Guizhou plateau, while the moderate- and high-risk areas decreased slightly (−5.51% and −0.88%) (Table 4; Figures 5–7).

The *A. adenophora* invasion-risk area exhibits a process of expansion followed by contraction under SSP585 climatic conditions, with a slight expansion in the total area. Between the present and the 2090s, the weed invasion-risk area has increased by $1.99 \times 10^4$ km$^2$, accounting for 3.55%, of which the expansion area is $6.60 \times 10^4$ km$^2$, accounting for 11.77%, and the total contraction area is $4.61 \times 10^4$ km$^2$, accounting for 8.22% (Table 5; Figures 6 and 7). The expansion area is in the Minjiang River and Jialing River in northern Sichuan, and the contraction area is in the Guizhou Province. In particular, the invasion-risk area of *A. adenophora* expanded slightly in the 2050s under SSPP585 climatic conditions in the southeast and northeast and in Taiwan, especially in the Minjiang River in Sichuan and in the Yungui Plateau region (Figures 6 and 7), and increased slightly northwards in the 2090s in the invasion-risk area in northern Sichuan (Figures 6 and 7). Overall, the risk area for the invasive spread of *A. adenophora* is more severe under the SSPP126 scenario than under the SSPP585 scenario.

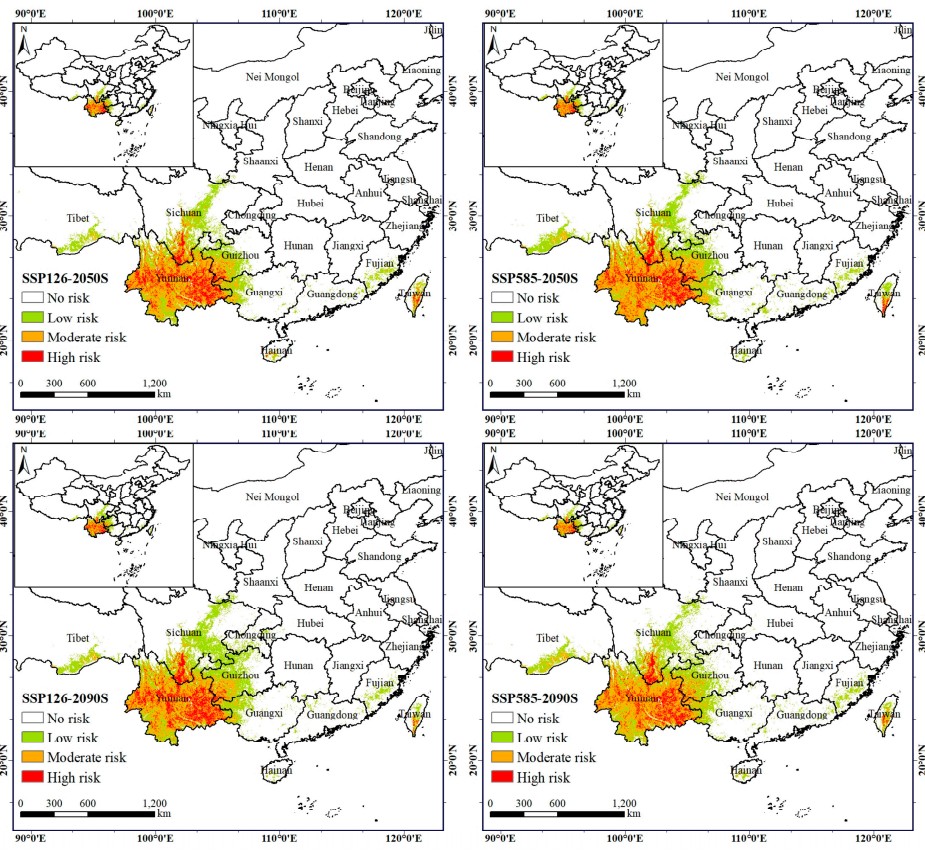

**Figure 6.** The spatial dispersal dynamics of *Ageratina adenophora* in the future.

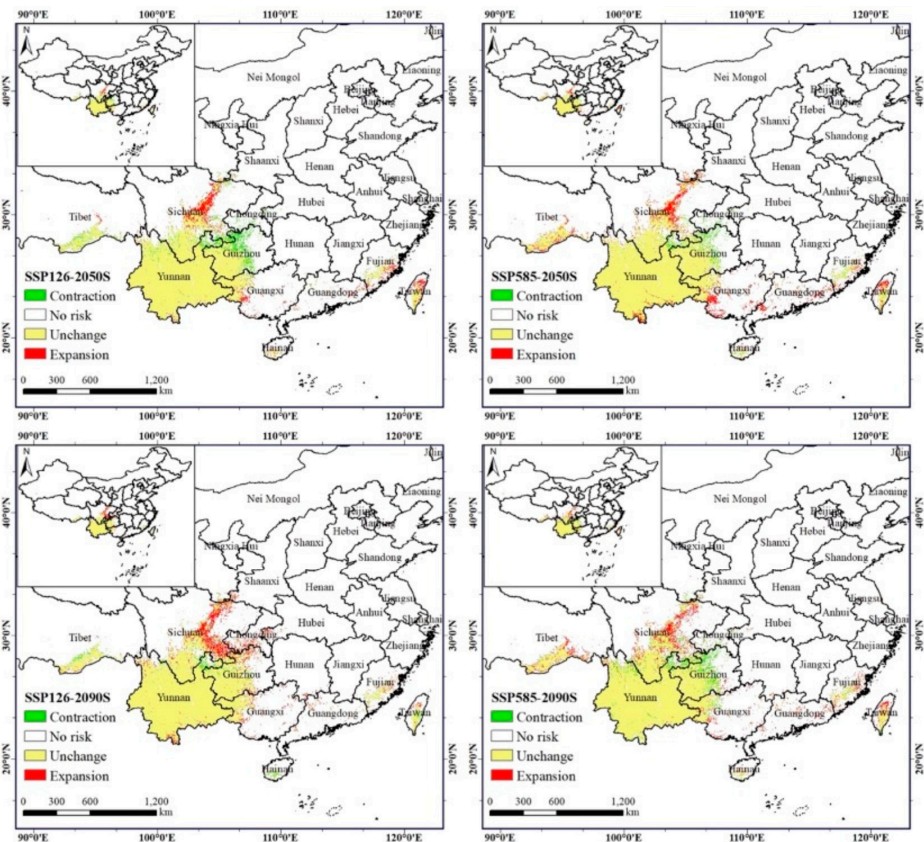

**Figure 7.** Spatial shifts patterns of dispersal risk for *Ageratina adenophora*.

**Table 5.** The area dynamics of *Ageratina adenophora* in the future.

| | Climate Scenarios | Area ($\times 10^4$ km$^2$) | | | | Area Ratio (%) | | | |
|---|---|---|---|---|---|---|---|---|---|
| | | Contraction | Unchanged | Expansion | Total Change | Contraction | Unchanged | Expansion | Total Change |
| 2050s | ssp126 | 5.32 | 50.79 | 4.75 | −0.57 | 9.49% | 90.51% | 8.46% | −1.02% |
| | ssp585 | 2.61 | 53.50 | 7.00 | 4.39 | 4.65% | 95.35% | 12.47% | 7.83% |
| 2090s | ssp126 | 1.93 | 54.17 | 7.52 | 5.59 | 3.45% | 96.55% | 13.40% | 9.96% |
| | ssp585 | 4.61 | 51.50 | 6.60 | 1.99 | 8.22% | 91.78% | 11.77% | 3.55% |

*3.5. The Centroid Distribution Transfer of the A. adenophora*

To determine the effects of climate change on the potential distribution area of *A. adenophora*, we used the spatial analysis function of ArcGIS to calculate the geometric center, migration distance, and direction of this alien invasive plant under different climatic scenarios (Figure 8). The centroid distribution of the *A. adenophora* under the current climate scenario is located in Kunming, Yunnan Province at (102.218° E, 25.095° N). The centroid distribution of the *A. adenophora* under the 2050s SSP126 scenario is located in Xundian Hui and Yi Autonomous County, Kunming, Yunnan Province at (103.257° E, 25.679° N), and has shifted to the northeast compared to its current position (Figure 8). The dispersal risk centroid distribution in the 2090s SSP126 scenario continued to move northeast compared to the 2050s position, reaching Huize County, Qujing City, Yunnan Province at (103.331° E, 26.017° N) (Figure 8), and the dispersal risk centroid distribution in the 2050s SSP585 scenario moved northeast and then southwest in the 2090s. In general, the dispersal risk centroid of this invasive plant moved all the way to the northeast and a relatively large distance under the SSP126 scenario, while it did not change much under the SSP585 climate scenario in 2081–2100.

Under the current period, the centroid of the high-risk zone is located in Lufeng County, Yunnan Province, where the centroid of the high-risk dispersal zone for the 2050s and 2090s under the SSP126 scenario migrates a small distance, generally to the southeast and then to the northwest. In the SSP585 scenario, the centroid migration pattern in the high-risk dispersal zone is similar to that in the SSP126 climate scenario, with a general migration to the southeast and then to the west (Figure 8). Overall, the high-risk dispersal zone centroid distribution of *A. adenophora* shows an overall trend towards the southeast from the present to the future.

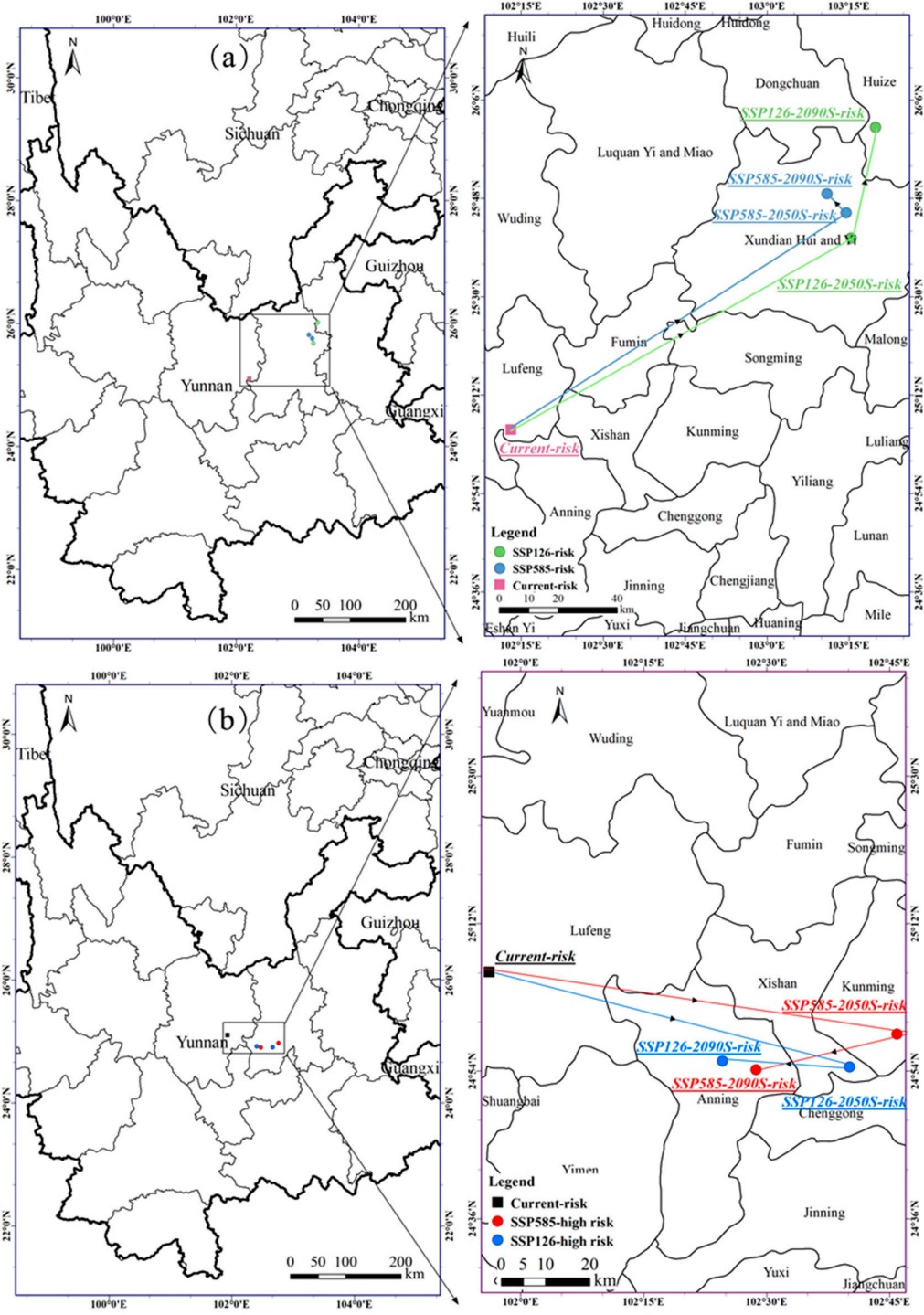

**Figure 8.** The *Ageratina adenophora* centroid transfer in the future periods (2041–2060, 2081–2100) for (**a**) the invasion total risk and (**b**) the invasion high-risk zones.

## 4. Discussion

The invasion of alien plants will threaten the stability of regional ecosystems and socioeconomic development. It is necessary to give early warning of alien invasive plants because early prevention of alien plants is thought to be a more effective economic strategy than control and elimination after the outbreak. *A. adenophora* is very aggressive in most parts of the world. Our research can provide the scientific basis for the prevention and control of alien invasive plants. Based on the distribution information of *A. adenophora* and data on climate, topography, soil, human activities, and land cover factors, this study used the MaxEnt model and ArcGIS spatial analysis techniques to construct the geographical distribution pattern of *A. adenophora* in the current and future periods, and to examine the key factors affecting the spread of the weed, so as to provide predictive and early warning information for the prevention and control of the invasive alien plant.

### 4.1. Potential Risk of Invasive Alien Plants under the Current and Future Scenario

The invasion-risk areas of *A. adenophora* in the current period are located along rivers and roads or in plains, mountains, and hills within the range of 18.394° N–33.653°N and 91.099° E–121.756° E. These regions include Yunnan, Sichuan, Tibet, Chongqing, Hubei, Guangxi, Guizhou, Fujian, and Taiwan. The climate in these regions is warm and humid, with plenty of sunshine and gentle terrain. The average annual temperature ranges from 13 °C to 22 °C, and the precipitation ranges from 800 to 1500 mm, decreasing from southeast to northwest [69,70]. *A. adenophora* distributes from south to north, and it is influenced by the high-altitude mountains in northwest Sichuan; the temperature of these areas is not suitable for the weed to survive, which prevents weed dispersal. This result is consistent with the findings of previous studies: *A. adenophora* grows best at an altitude around 2000 m, but once above 2500 m, it becomes difficult for the weed to spread [21,71,72]. Thus, the northernmost point is concentrated along the Minjiang and Jialing Rivers, and the southernmost point to the Lancang river basin in the Yunnan section.

Our predictions show that *A. adenophora* invades southern China, with Yunnan, western Guangxi, western Guizhou, and southwestern Sichuan being the most abundant, especially along rivers and roads, and a small area is also invaded in Tibet, Chongqing, Hubei, Guangzhou, Fujian, and Taiwan. The climate in southern China is tropical and subtropical monsoonal, with hot and rainy summers and mild and dry winters in most areas, and the terrain is dominated by plains, mountains, and hills, which are very similar to the natural conditions in the area of the origin of *A. adenophora* [73]. Mexico is a highland mountainous country with an average annual temperature of 17–27 °C and annual precipitation of 750–2000 mm. The other country of origin, Costa Rica, has an average annual temperature of 16–25 °C and annual precipitation of 2540 mm [71]. Therefore, after spreading from Myanmar to China, this weed colonized the Yunnan–Guizhou plateau and the Sichuan basin and dispersed to the coastal zone of China.

Compared to its current distribution area, *A. adenophora* is likely to continue to spread in northeast Sichuan in the future, but its spread will probably slow down, and its distribution area may contract to some extent near the Yunnan–Guizhou Plateau. From the predictions, there is still a spread risk in the Sichuan Basin, the Yunnan–Guizhou Plateau, Guangxi, Tibet, Gansu, Shaanxi, Chongqing, Hubei, Taiwan, and Fujian. This result suggests that the current distribution of *A. adenophora* may not reach the maximum suitable invasion area under the SSP126 emission scenario and will reach the maximum invasion range under the 2050s-SSP585 emission scenario. One possibility is related to global climate change in recent years, which has contributed to the increase in atmospheric temperature and the frequent occurrence of extreme weather [74], which may limit the spread of this weed. The other possibility is that extreme weather causes the model to be reluctant to make strong predictions in order to avoid exaggerating the results.

The high-risk invasion areas of this weed are mainly located in Yunnan Province and the western part of Sichuan Province, which is consistent with the phenomena observed during our field surveys and the results of previous studies [27]. Yunnan Province and the

Panxi region have ideal climatic and topographical conditions for the growth of this weed. Roads like the Rongli Highway and the Xuwei Highway have created ideal conditions for the spread of this invasive plant. As a result, Yunnan Province's flat topography and the dense water flow roads in the Panxi region are high-risk areas for weed invasion.

### 4.2. Key Drivers Affecting Invasive Alien Plants

Most studies on SDM have only selected climate variables to be involved in modelling and prediction [75,76]. In this study, climate, land cover, soil, topography, and human activity factors were all taken into account. Based on the results of the contribution of the variables in this study, the key drivers influencing the invasion of *A. adenophora* were screened, and we can further understand the environmental conditions for the invasion of this weed. We found that the dominant factors influencing the distribution pattern of *A. adenophora* were temperature annual range (bio_7), elevation (dem), mean annual precipitation (bio_12), mean temperature of driest quarter (bio_9), and human activity (hii).

Both temperature and precipitation play an important role in the formation of invasive alien plant dispersal habitats, and our results are consistent with previous studies [70]. Altitude has a significant inhibitory effect on the invasion of this weed, and our results show that *A. adenophora* spreads in the northeast, a distribution pattern caused by the presence of a large number of high-altitude mountains (e.g., Gonggar Mountain, Minshan mountains, and Daba Mountain) in northwestern Sichuan preventing the continued spread of the invasive alien plant. It is worth noting that although human activities can contribute to some extent to the spread of invasive alien species, we found that human activities had a smaller impact on this invasive weed compared with temperature, precipitation, and topographic factors, with a contribution of only 7.8%. However, human activity had a higher impact on this weed than the soil variable (single-factor contribution less than 0.7%, cumulative contribution of 2.8%) and the land cover variable (contribution of 0.1%) (Table 3). This is because the roots of *A. adenophora* have an allelopathic effect, releasing harmful substances into the soil environment and thus inhibiting the growth of other plants [20]. Land use/cover change does not significantly affect the distribution of *A. adenophora*. According to our extensive field surveys and herbarium records (China Digital Herbarium), *A. adenophora* can be distributed in open forests, roadsides, abandoned fields, ridge edges, riversides, orchards, etc. As long as the temperature, moisture, and sunlight are suitable, the weed can colonize and spread. Overall, *A. adenophora* has low requirements for land cover, soil type, soil texture, and soil quality, and, under suitable climatic conditions, increased human activity can contribute to the rate and extent of spread of invasive alien species in invaded areas [77]. It is therefore particularly important to control the impact of human activities on invasive alien plants during the spread of invasive plants.

### 4.3. Uncertainty

The limitations of this study can be summarized as follows. Since MaxEnt is a species distribution model, its niche conservation premise may cause MaxEnt model to be reluctant to make strong predictions in some places with small sample sizes or extreme environmental conditions, which may explain why the distribution area of *A. adenophora* is reduced under extreme climate scenarios. Our species distribution dataset comes from open datasets and field sampling. Although we tried to be as uniform and unbiased as possible, this does not ensure that there was no bias at all.

## 5. Conclusions

Predicting the distribution area of alien invasive plants under climate change is crucial to the early monitoring of the regional ecological environment and biodiversity conservation. We predicted the spatial and temporal distribution patterns and dispersal areas of *A. adenophora* under different climate scenarios. Under the current climate scenario, *A. adenophora* has invaded several provinces in southwestern and southern China, mainly

Yunnan, Sichuan, Guizhou, Chongqing, Tibet, Hubei, and coastal areas (Guangxi, Fujian, Taiwan, etc.), which provide a warm and humid climatic environment for this invasive plant. The future distribution area will probably keep dispersing in the northeast and along the coastal areas. Therefore, we should strengthen the monitoring and management of these areas to prevent damage to the regional economy and ecology caused by the alien invasive plants. For the invaded area, considering the high cost and inefficiency of manual and mechanical control, and the limitation of chemical control to cause environmental pollution, biological and ecological control techniques should be adopted to prevent its colonization and dispersal, especially in potential distribution areas. The key drivers affecting the distribution of *A. adenophora* are: temperature annual range (bio_7), elevation (dem), mean annual precipitation (bio_12), mean temperature of driest quarter (bio_9), and human activity (hii). Therefore, while monitoring and managing these potential distribution areas, we should also strengthen the management of human activities to prevent the spread of invasive plants. Our study provides fundamental information for the early detection of and rapid response to alien invasive species in a new habitat, helps to determine the future invasion area of invasive plants, and proposes preventive measures. We suggest the prevention of invasive plant outbreaks before they occur, rather than elimination or more expensive economic strategies afterward.

**Supplementary Materials:** The following are available online at https://www.mdpi.com/article/10.3390/d14110915/s1, Table S1: Environmental variables.

**Author Contributions:** Conceptualization, X.Z., Y.W. and P.P.; Methodology, X.Z. and Z.T.; Software, X.Z.; Validation, X.Z., Y.W., G.W. and P.P.; Formal Analysis, G.W. and G.Z.; Writing—Original Draft Preparation, X.Z.; Writing—Review & Editing, X.Z., Y.Z., Y.W., P.P. and G.W.; Visualization, X.Z. and Y.Z.; Funding Acquisition, P.P. and G.W. All authors have read and agreed to the published version of the manuscript.

**Funding:** This work was funded by the Second Tibetan Plateau Scientific Expedition and Research Program of P. R. China (2019QZKK0301) and the National Natural Science Foundation of P. R. China (31860123 and 31560153).

**Institutional Review Board Statement:** Not applicable.

**Data Availability Statement:** Publicly available datasets were analyzed in this study. This data can be found here: https://www.gbif.org; https://www.cvh.ac.cn; https://sedac.ciesin.columbia.edu/data/set/wildareas-v2-human-influence-index-geographic; http://www.worldclim.org; http://www.resdc.cn; http://www.nsii.org.cn/2017/home.php; http://www.fao.org/soils-portal/soil-survey/soil-maps-and-databases/harmonized-world-soil-database-v12/en/ (all accessed on 28 April 2022).

**Acknowledgments:** We thank the editors and anonymous experts for their comments on this study.

**Conflicts of Interest:** The authors declare no conflict of interest.

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
