# Peer review of "Mapping the Distribution and Dispersal Risks of the Alien Invasive Plant Ageratina adenophora in China"

_diversity, doi:10.3390/d14110915_

Round 1

Reviewer 1 Report

The authors have presented an interesting and well-written study of the potential distribution of the invasive exotic plant Ageratina adenophora in China under future climate scenarios. While the use of species distribution models (SDM) such as MaxEnt to predict the potential distribution of invasive species is not a new approach, this is a well designed and conducted study and provides a model for other investigators to follow.

The introduction provides a thorough overview of the problems posed by A. adenophora invasion and the likelihood of continued spread. The use of species distribution models such as MaxEnt is clearly explained and justified; however, more clarity is required regarding the use of CIMP5 and CIMP6. In particular explain how these interact with and inform the MaxEnt modelling. 

The methods are detailed and robust. Of particular note is that the authors have included in the model a 18 environmental factors that may influence distribution. These include include temperature, elevation, slope, soil type and human activity. As noted in the discussion, most SDM studies have only selected climate variables to make predictions. This study thus provides a worthwhile example to other investigators of an approach that provides for greater precision in SDM. 

The results are clear and well presented with the key drivers of invasion and future distribution. Several climate scenarios are modelled and detailed information regarding likely distribution under those scenarios is provided. This is of value in managing the threat as the most important factors can be prioritised for management. It is an approach that has wider applicability to the management of invasive species. 

Conclusions follow logically from the results; however, the discussion could be strengthened by exploring the wider applicability of the approach used in this study.

There are some very minor grammatical errors but these can easily be corrected in final editing. 

Author Response

Dear editors, and reviewers,

On behalf of my co-authors, we thank you very much for giving us an opportunity to revise our manuscript, we appreciate editor and reviewers very much for their positive and constructive comments and suggestions on our original manuscript entitled “Mapping the distribution and dispersal risks of the alien invasive plant Ageratina adenophora in China”. (Manuscript Number: diversity-1886432).

We have studied the reviewer’s comments carefully and have made revisions marked up using the “Track Changes” function in the manuscript. We have tried our best to revise our manuscript according to the comments. Attached please find the revised version, which we would like to submit for your kind consideration. Below you will find our point-by-point responses to the reviewers’ comments/questions:

Responds to the editor and reviewer’s comments:

Response to Reviewer 1 Comments

The authors have presented an interesting and well-written study of the potential distribution of the invasive exotic plant Ageratina adenophora in China under future climate scenarios. While the use of species distribution models (SDM) such as MaxEnt to predict the potential distribution of invasive species is not a new approach, this is a well designed and conducted study and provides a model for other investigators to follow.

Point 1: The introduction provides a thorough overview of the problems posed by A. adenophora invasion and the likelihood of continued spread. The use of species distribution models such as MaxEnt is clearly explained and justified; however, more clarity is required regarding the use of CIMP5 and CIMP6. In particular explain how these interact with and inform the MaxEnt modelling. 

Response 1: Thanks for your comments. We have added the literature on the comparison between CMIP5 and CMIP6 to illustrate the advantages of CIMP6 over CIMP5, and also added the research on species distribution prediction based on CIMP5 using maxent. In addition, in Section 2.2 para 3, we introduced the shared socio-economic pathways in the CMIP6 model we used.

Modification 1: (Introduction para 2) “In a comparison of the performance of CMIP6 and CMIP5 models in simulating historical precipitation and temperature in Bangladesh, Kamruzzaman et al found that the CMIP6 MME showed a significant improvement in simulating rainfall and temperature over Bangladesh compared to CMIP5 MME (Kamruzzaman et al. 2021).  Chen et al. showed that the new version of CMIP6 models exhibits a general improvement in the simulation of climate extremes and their trend patterns compared to the old version of the model in CMIP5, and the simulation results are closer to the observations (Chen et al. 2020). In addition, an increasing number of researchers have recently used CMIP6 models for species distribution prediction (Gao et al. 2021; Ramasamy et al. 2022).”

(Section 2.2 para 3) “We selected the BCC-CSM2-MR global climate model from the coupled phase 6 (CMIP6) model for our projections because of its higher resolution and climate sensitivity compared to CMIP5 (Hamed et al. 2022; Petrie et al. 2021). and the better performance of the BCC-CSM2-MR model compared to other global climate models (Wu et al. 2021). For the four shared socio-economic pathways (SSP126, SSP245, SSP370 and SSP585), we chose SSP126 for the optimistic scenario and SSP 585 for the pessimistic scenario for model simulation under the future climate scenario, as they combine factors affecting local development and a higher climate sensitivity (Meinshausen et al. 2020).”

The methods are detailed and robust. Of particular note is that the authors have included in the model a 18 environmental factors that may influence distribution. These include temperature, elevation, slope, soil type and human activity. As noted in the discussion, most SDM studies have only selected climate variables to make predictions. This study thus provides a worthwhile example to other investigators of an approach that provides for greater precision in SDM. 

The results are clear and well presented with the key drivers of invasion and future distribution. Several climate scenarios are modelled and detailed information regarding likely distribution under those scenarios is provided. This is of value in managing the threat as the most important factors can be prioritised for management. It is an approach that has wider applicability to the management of invasive species. 

Point 2: Conclusions follow logically from the results; however, the discussion could be strengthened by exploring the wider applicability of the approach used in this study.

Response 2: Thanks for your suggestion. We agree with you, and we have supplemented the wider adaptability analysis of our research in this field.

Modification 2: (Discussion para 1) “The invasion of alien plants will threaten the stability of regional ecosystems and socio-economic development. It is necessary to give early warning to alien invasive plants because early prevention of alien plants is thought to be a more effective economic strategy than control and elimination after the outbreak. A. adenophora is very aggressive in most parts of the world, our research can provide the scientific basis for the prevention and control of alien invasive plants. Based on the distribution information of A. adenophora and data on climate, topography, soil, human activities and land cover factors, this study used the MaxEnt model and ArcGIS spatial analysis techniques to construct the geographical distribution pattern of A. adenophora in the current and future periods, and to examine the key factors affecting the spread of the weed, so as to provide predictive and early warning information for the prevention and control of the invasive alien plant.”

(Conclusions) “Predicting the distribution area of alien invasive plants under climate change is crucial to the early monitoring of the regional ecological environment and biodiversity conservation. We predicted the spatial and temporal distribution patterns and dispersal areas of A. adenophora under different climate scenarios. Under the current climate scenario, A. adenophora has invaded several provinces in southwestern and southern China, mainly including Yunnan, Sichuan, Guizhou, Chongqing, Tibet, Hubei and coastal areas (Guangxi, Fujian, Taiwan, etc.), which provide a warm and humid climatic environment for this invasive plant. The future distribution area probably keep dispersing in the northeastern and along the coastal areas. Therefore, we should strengthen the monitoring and management of these areas to prevent damage to the regional economy and ecology caused by the alien invasive plants. For the invaded area, considering the high cost and inefficiency of manual and mechanical control, and the limitation of chemical control to cause environmental pollution, biological and ecological control techniques should be adopted to prevent its colonization and dispersal, especially in potentially distribution areas. The key drivers affecting the distribution of A. adenophora are: Temperature Annual Range (bio_7), elevation (dem), Mean Annual Precipitation (bio_12), Mean Temperature of Driest Quarter (bio_9) and human activity (hii). Therefore, while monitoring and managing these potential distribution areas, we should also strengthen the management of human activities to prevent the spread of invasive plants. Our study provides the fundamental information for early detection and rapid response to alien invasive species in a new habitat, helps to determine the future spread area of invasive plants, and proposes preventive measures. We suggest the prevention of invasive plant outbreaks before they occur, rather than elimination or more expensive economic strategies afterward.”

Point 3: There are some very minor grammatical errors but these can easily be corrected in final editing. 

Response 3: Thank you for pointing out our mistakes. We have checked the whole manuscript for grammatical errors.

REFERENCES

Chen H, Sun J, Lin W, and Xu H. 2020. Comparison of CMIP6 and CMIP5 models in simulating climate extremes. Science Bulletin 65:1415-1418. 10.1016/j.scib.2020.05.015

Gao T, Xu Q, Liu Y, Zhao J, and Shi J. 2021. Predicting the potential geographic distribution of Sirex nitobei in China under climate change using maximum entropy model. Forests 12:151.

Hamed MM, Nashwan MS, Shahid S, bin Ismail T, Wang X-j, Dewan A, and Asaduzzaman M. 2022. Inconsistency in historical simulations and future projections of temperature and rainfall: A comparison of CMIP5 and CMIP6 models over Southeast Asia. Atmospheric Research 265:105927.

Kamruzzaman M, Shahid S, Islam A, Hwang S, Cho J, Zaman M, Uz A, Ahmed M, Rahman M, and Hossain M. 2021. Comparison of CMIP6 and CMIP5 model performance in simulating historical precipitation and temperature in Bangladesh: a preliminary study. Theoretical and Applied Climatology 145:1385-1406.

Meinshausen M, Nicholls ZR, Lewis J, Gidden MJ, Vogel E, Freund M, Beyerle U, Gessner C, Nauels A, and Bauer N. 2020. The shared socio-economic pathway (SSP) greenhouse gas concentrations and their extensions to 2500. Geoscientific Model Development 13:3571-3605.

Petrie R, Denvil S, Ames S, Levavasseur G, Fiore S, Allen C, Antonio F, Berger K, Bretonnière P-A, and Cinquini L. 2021. Coordinating an operational data distribution network for CMIP6 data. Geoscientific Model Development 14:629-644.

Ramasamy M, Das B, and Ramesh R. 2022. Predicting climate change impacts on potential worldwide distribution of fall armyworm based on cmip6 projections. Journal of Pest Science 95:841-854.

Wu T, Yu R, Lu Y, Jie W, Fang Y, Zhang J, Zhang L, Xin X, Li L, and Wang Z. 2021. BCC-CSM2-HR: a high-resolution version of the Beijing Climate Center Climate System Model. Geoscientific Model Development 14:2977-3006.

Other modifications

In addition, we have also actively revised some contents as follows :

  1. We have corrected the grammatical errors using the “Track Changes” function in the revised manuscript.
  2. We have checked the whole manuscript and corrected the misspelling.
  3. We have changed the abbreviations in Table 1 and Appendix 1 into lower case letters to unify the expression in the text, and checked the format of all tables in this manuscript.
  4. We have revised our manuscript according to the comments of other reviewers.

We have tried our best to improve the manuscript and made some changes in the manuscript. These changes will not influence the content and framework of the paper. We appreciate for Editors/Reviewers’ warm work earnestly, and hope that the correction will meet with approval.

We would like to express our great appreciation to you and reviewers for comments on our manuscript.

Thank you and best regards.

Yours sincerely,

Xiaojuan Zhang

[email protected]

Institute of Ecological Resources and Landscape Architecture, Chengdu University of Technology

Reviewer 2 Report

Thank you for the opportunity to review this paper, which uses MAXENT to predict the current and future distribution of an invasive plant in China.

Overall the work is substantial, competently done and well presented (with some English editing required). The topic was of interest to me, and I imagine to other potential readers.

However, I have two significant reservations:

1) The work- although competent- is not particularly novel. All of the work is conducted using existing off-the-shelf software packages, and the ideas presented are not new. I see Diversity as a good journal which should require innovation in thought or method, and I think that level of innovation is lacking in this paper (or, at least, it is under-sold).

2) I have serious reservations about the underlying concept of modelling future distributions of invasive species based on their current distributions. My central concern is that there is no evidence that the species has fully occupied its potential niche within China, and therefore its capacity to tolerate conditions beyond those it currently experiences is simply unknown (As a thought experiment: If we performed the same project only 1 year after first invasion of the weed, the distribution would be small, and the model projections would only predict occupancy of small areas similar to those first occupied. If we repeated the process 50 years after introduction when the same species had invaded further, the result would be very different). 

Point 2 is, unfortunately, very difficult to address. I see two ways to improve the paper:

The first is to discuss the issues of prediction for species not yet at equilibrium more deeply. Currently, the authors seem overly confident that their models are good at extrapolating. I am not so sure.

The second is to examine and model using the native range of the species in Mexico, and see whether the models differ in their delineation of niche, and whether the models can cross-predict each other, or whether they differ a lot. If they show similar niche parameters, and cross-predict each other closely, this is an argument that the species in China is indeed controlled by the predictors in the model, and that the projected distributions are more likely to be correct. If not- there is much more to talk about!

If this paper did both of these things, it would be very much improved in my opinion. Given these are major changes requiring new work, I have ticked "major revision".

In addition to these major concerns, I noted the following moderate - minor issues:

Throughout- minor English expression needs review.

Introduction para 2: The species is not yet mentioned! It must be introduced by name at this point. In fact, the genus is never mentioned anywhere in the introduction.

Introduction para 2: "chemosensory". I don't think this is the correct word. It is commonly used to refer to taste/smell in animals. I think maybe "allelopathic" is what is meant?

Introduction para 3: The introductory statement about the use of models should be qualified a little- their utility is still debated.

Introduction para 3: "...in the absence of species distribution data..." requires clarification / explanation.

Introduction last para: ArcGIS is not a technique.

Methods para 1: "...each species..." What does this mean? Only one species is examined.

Methods para 1: No records are visible in Tibet or Hubei as claimed in text.

Results: throughout: Many claims are made with too much confidence (e.g. ...was able to fully reflect the distribution..."), and must be reworded to introduce some nuance and appropriate uncertainty. Similarly, most of the numbers are over-precise, given the context.

Section 3.3 para 2: The association with the road network is likely to reflect collection bias (near roads) as much as the ecology of the species (although it may be ecologically associated with roads- who can tell). This reveals some naivety in the presentation and discussion of the results.

Figure 4: The phrase "current potential" is confusing. I think a better term would be "Risks under current climate" or something.

Section 3.4: The fact that future climate predict some contraction of range: could this be because future projections include novel / unsampled combinations of climate and geography, and therefore the model is reluctant to predict strongly in these areas? I do not understand MAXENT well enough to know, but this seems possible to me, and is a point worth discussion.

Section 3.5 seems completely pointless to me. Why does the centroid of the distribution matter? I can't see why anyone would want to know this. I think this section can be deleted (unless I am missing the point- in which case the authors should explain why this is of interest and ecological meaning).

Section 4.1: The discussion about the weed being unable to over-top mountain ranges needs more nuance. Is there suitable habitat on the other side of the mountain? If so, the future is almost completely unpredictable: the weed may never cross, but a fluke event mediated by people could spread it there very soon (just as the species arrived in China).

Section 4.1: "Therefore the weed is rapidly adapting..." This phrase is deeply problematic. First, the results presented say absolutely nothing about adaptation. Second, if the species is indeed adapting to new conditions, then a correlative modelling approach to define its future distribution seems virtually pointless. 

Section 4.1: Phrases like "...its spread will slow down.." are far too confident and deterministic. The paper needs to be worded more cautiously, with more acknowledgement of ecological and model-derived uncertainty.

Conclusion: "We have successfully simulated and predicted..." This is too strong. You can not test your future predictions- no one yet knows how successful this work has been in predicting. Please re-word.  

Reviewer 3 Report

See the attached

Round 2

Reviewer 2 Report

Thank you for allowing me to look at this paper again.

I have read the revision, and considered the response to each of my previous comments.

I am not convinced that this paper has improved significantly. Regarding novelty, the two points offered still don't convince me that the paper is particularly novel. Inclusion of a few more predictor variables (terrain, human activity, etc) does not, in my mind, add substantial novelty. Nor does prediction under future scenarios. Both of these things have been done many times before (although it is true that many papers don;t do them).

I have thought about what the examination of the native range in Mexico offers. I don't think this is enough. Just because the same variables are strong predictors in each case does not show that the distributions are 'stable' / 'realised' cross-predictable. I can easily imagine cases where a variable is a strong / important predictor in two models, but the outcomes of the predictions are quite different. I really think that a proper cross-prediction is required here.

The minor changes have been well made, thank you.

Overall, I feel that the paper is still not up to the standard of novelty and thoughtfulness that Diversity would seem to require.
